# Probabilistic Framework for Robustness of Counterfactual Explanations Under Data Shifts

**Xuan Zhao**[1*] **Lena Krieger**[1,2*] **Zhuo Cao**[1] **Arya Bangun**[1] **Hanno Scharr**[1] **Ira Assent**[1,3]

[1]IAS-8, Forschungszentrum Jülich, Germany
[2]Munich Center for Machine Learning (MCML), LMU Munich, Germany
[3]Department of Computer Science, Aarhus University, Denmark
`{xu.zhao, l.krieger, z.cao, a.bangun, h.scharr, i.assent}@fz-juelich.de`

## Abstract

Counterfactual explanations (CEs) are a powerful method for interpreting machine learning models, but CEs might be not valid when the model is updated due to distribution shifts in the underlying data. Existing approaches to robust CEs often impose explicit bounds on model parameters to ensure stability, but such bounds can be difficult to estimate and overly restrictive in practice. In this work, we propose a data shift-driven probabilistic framework for robust counterfactual explanations with plausible data shift modeling via a Wasserstein ball. We formalize a linearized Wasserstein perturbation scheme that captures realistic distributional changes which enables Monte Carlo estimation of CE robustness probabilities with domain-specific data shift tolerances. Theoretical analysis reveals that our framework is equivalent in spirit to model parameter bounding approaches but offers greater flexibility, avoids the need to estimate maximal model parameter shifts. Experiments on real-world datasets demonstrate that the proposed method maintains high robustness of CEs under plausible distribution shifts, outperforming conventional parameter-bounding techniques in both validity and proximity costs.

## 1 Introduction

Counterfactual explanations (CEs) have emerged as a crucial tool for interpreting machine learning (ML) models by offering insights into examples of changed input features leading to a different prediction [Wachter et al., 2017]. In applications such as finance, healthcare, and hiring, these explanations provide actionable feedback, enabling users to understand and potentially influence automated decisions [Karimi et al., 2020, Ustun et al., 2019]. However, a critical challenge limiting the practicality of CEs is their lack of robustness to data shifts. As real-world data distributions evolve over time due to factors such as demographic changes, policy updates, or external perturbations [Blodgett et al., 2016, Beery et al., 2021], CEs that were once valid may become obsolete or misleading. This fragility undermines the trustworthiness and utility of CEs in dynamic environments.

Consider a loan application scenario: if a model is retrained while an applicant is actively improving their credit profile, a non-robust CE may still indicate approval, despite the updated features warranting rejection. Such inconsistencies could expose institutions to liability, as prior guidance may conflict with the revised outcome. Ensuring that a CE remains valid after model updates is thus critical for maintaining actionable and reliable explanations.

Existing approaches to CEs primarily focus on optimizing criteria such as sparsity, proximity, and feasibility [Wachter et al., 2017, Verma et al., 2020, Karimi et al., 2021]. While some recent works

---

*These authors contributed equally.

39th Conference on Neural Information Processing Systems (NeurIPS 2025) Workshop: Reliable ML from Unreliable Data.

attempt to incorporate robustness, they often fail to accurately model realistic data shifts that occur in practice, even though some assume only minor perturbations to the data. Consequently, these methods struggle to provide meaningful guarantees on the reliability of CEs in real-world deployments.

In this work, we propose a novel framework that explicitly models possible future data shifts to ensure the robustness of CEs. Rather than relying on generic worst-case perturbations or static assumptions about distributional changes, we adopt a structured and flexible approach based on Wasserstein-bounded data shifts. By linking plausible data perturbations to bounded model parameter changes under fine-tuning, our method allows for probabilistic guarantees on the validity of CEs. This approach not only maintains robustness but also preserves lower proximity, providing a practical and principled solution for reliable CEs.

**Related Works**  *Robustness in CEs* has been explored from multiple angles, including robustness against noisy execution, input variations, model multiplicity, and, what we focus on, model changes. For a comprehensive overview, we refer to a recent survey [Jiang et al., 2024].

*Model parameter updates* are usually triggered by retraining on new data [Rawal et al., 2020]. Recent works explore different types of boundaries when assessing model changes, including explicit bounds, data shifts, and parameter shifts. Some approaches generate CEs that achieve high certainty, i.e., high class scores, to ensure robustness [Dutta et al., 2022, Hamman et al., 2023, Jiang et al., 2023], while others augment data with CEs in training [Ferrario and Loi, 2022], or perturb the covariance matrix of data sampled close to the decision boundary to train a linear surrogate [Bui et al., 2022].

Upadhyay et al. [2021] introduce plausible model shifts, constraining parameter updates to perturbations within a predefined magnitude. Their method ROAR, minimizes the loss over the worst plausible model shift to generate CEs. Thereby the possibly nonlinear model is approximated with a linear surrogate. AP$\Delta$S [Marzari et al., 2024] samples plausible model shifts, unlike our approach where we sample data distributions. MILP [Mohammadi et al., 2021] is based on linearly approximating ReLu activated neural networks to compute bounds on the hidden units employing Mixed-Integer-Programming. A key limitation to these approaches is their assumption that model changes stem from minor data shifts, without assessing data shifts explicitly. Nguyen et al. [2022] assess model shifts by adversarially perturbing the data using a Gaussian mixture ambiguity set and Wasserstein distance. The data distribution is hereby modeled with kernel density estimation, which is prone to suffer from the curse of dimensionality. Our method proactively models the probable data shifts. Unlike most works, ours is a model-agnostic post-hoc approach that can be employed to enhance robustness in existing CE generation methods.

*Wasserstein distributionally robust optimization* (WDRO) allows decision-making in uncertain cases by accounting for an initial distribution and distributions within its neighborhood, measured with a Wasserstein ball [Kuhn et al., 2019]. The Wasserstein distance, from optimal transport, compares probability distributions by considering the geometry of the underlying space. It measures the cost of transporting one distribution to another. WDRO has gained popularity in different ML disciplines, including adversarial training [Liu et al., 2025], causality and fairness [Ehyaei et al., 2024].

## 2   Robust Counterfactuals under Wasserstein Perturbations

**Problem Setup**  Let $\hat{\mathbb{P}}_{\text{old}}$ denote the empirical distribution of historical data, and let $f_{\theta_{\text{old}}}$ be a pre-trained classifier. We denote the score (logit) for a given input $x \in \mathcal{X}$ $f_{\theta_{\text{old}}}(x) = y^m$, with $\arg\max y^m = c$ denoting the class, $m$ is the number of classes. For a given input $x \in \mathcal{X}$ with $\arg\max f_{\theta_{\text{old}}}(x) \neq c_{\text{target}}$ and a desired target class $c_{\text{target}}$, a counterfactual $x'$ is an alternative input such that $\arg\max f_{\theta_{\text{old}}}(x') = c_{\text{target}}$. We aim to ensure that $x'$ remains valid even under plausible future data shifts. To formalize this, we define a Wasserstein ball around $\hat{\mathbb{P}}_{\text{old}}$: $\mathcal{B}_{\varepsilon_x}(\hat{\mathbb{P}}_{\text{old}}) = \{\mathbb{P}_{\text{new}} : W_p(\mathbb{P}_{\text{new}}, \hat{\mathbb{P}}_{\text{old}}) \leq \varepsilon_x\}$, where $W_p$ is the Wasserstein distance with cost $c(\cdot, \cdot)$, and $\varepsilon_x$ is a radius determined from historical data or domain knowledge. For $p = 2$, $W_p$ is the *2-Wasserstein distance*. The Wasserstein distance is especially relevant for counterfactuals as these aim to transport a query sample toward the decision boundary. For each plausible data distribution $\mathbb{P}_{\text{new}} \in \mathcal{B}_{\varepsilon_x}(\hat{\mathbb{P}}_{\text{old}})$, fine-tuning the model yields updated parameters: $\theta_{\text{new}} = \arg\min_\theta \mathbb{E}_{x \sim \mathbb{P}_{\text{new}}}[\ell(x; \theta)]$, where $\ell(x; \theta)$ is the classification loss (cross-entropy in our experiments) of a model $f$. This defines a corresponding parameter set: $\Theta_\varepsilon = \{\theta_{\text{new}} : \mathbb{P}_{\text{new}} \in \mathcal{B}_{\varepsilon_x}(\hat{\mathbb{P}}_{\text{old}})\}$.

## Robust Counterfactuals under Wasserstein Perturbations

*Linearized Wasserstein Perturbation.* For small shifts, we approximate $\mathbb{P}_{\text{new}}$ via a linearized perturbation of each data point: $x_j^{\text{new}} = x_j + \delta_j, j = 1, \ldots, n$, subject to the linearized Wasserstein constraint: $\frac{1}{n} \sum_{j=1}^{n} d(x_j, x_j + \delta_j) \leq \varepsilon_x$. This first-order approximation allows efficient sampling of plausible future data shifts while remaining within the Wasserstein ball.

*Bounding Model Parameter Shifts under Wasserstein Perturbations.* We can formally show that the model parameter shift induced by a data perturbation within a Wasserstein ball is bounded. The full proof can be found in the Appendix A.1. Here, we briefly present main idea. To derive an upper bound under the given assumptions, we first derive the basic stability inequality including $\theta_{old}$ and $\theta_{new}$. Rearranging this inequality and making use of the triangle inequality and Lipschitz assumption allows us to derive the upper bound using the Cauchy–Schwarz inequality and linearized Wasserstein inequality. The boundedness of the model parameters favors the use of Monte Carlo sampling.

*Monte Carlo Sampling and Fine-Tuning.* To evaluate robustness, we generate $N$ Monte Carlo samples of perturbations $\{\delta_j^{(i)}\}_{j=1}^{n}$, with $i = 1, \ldots, N$. For each perturbed dataset, we fine-tune the model parameters $\theta_{\text{old}}$ by performing one or a few gradient steps:

$$\theta_{\text{new}}^{(i)} = \theta_{\text{old}} - \eta \, \nabla_\theta \left( \frac{1}{n} \sum_{j=1}^{n} \ell\big(f_\theta(x_j + \delta_j^{(i)}), y_j\big) \right),$$

where $\ell$ is the model loss function and $\eta$ is the learning rate. This procedure yields $N$ perturbed models $\{\theta_{\text{new}}^{(i)}\}_{i=1}^{N}$, representing plausible shifts in the decision boundary.

*Counterfactual Robustness Estimation.* We formalize robustness under sampled model shifts using the following lemma [Marzari et al., 2024, Wilks, 1942]:

**Lemma 2.1.** *Fix an integer $N > 0$ and a robustness threshold $R \in (0, 1)$. Suppose a candidate counterfactual $x'$ in class $c$ ($\arg\max f_{\theta_{old}}(x') = c$) is evaluated under $N$ independently sampled perturbed models $\Delta_N = \{\theta_{new}^{(i)}\}_{i=1}^{N}$ from the distribution of preturbed models $\Delta$. Then, with probability at least $\alpha = 1 - R^N$ over the sampling, we have:*

$$\mathbb{P}_{\theta_{new} \sim \Delta} \left( \min((f_{\theta_{new}}(x'))_c) \geq \min((f_{\Delta_N}(x'))_c) \right) \geq R \tag{1}$$

Informally, Lemma 2.1 enables determination of the minimum number of Monte Carlo samples $N$ required to guarantee that $x'$ is robust for at least a fraction $R$ of distributional shifts with probability $\alpha$. For example, setting $\alpha = 0.999$ and $R = 0.995$ yields $N = \log_R(1 - \alpha) \approx 1378$. If the lower bound of the reachable set, computed as $\min f_{\Delta_N}(x')_c$, exceeds a decision threshold (e.g., 0.5), we can assert with confidence $\alpha = 0.999$ that $x'$ remains valid for at least $R = 0.995$ of plausible data shifts.

*Summary* This framework leverages *linearized Wasserstein perturbations* to generate plausible future data shifts, fine-tunes the model efficiently, and uses Monte Carlo estimation to quantify counterfactual robustness. The method balances computational tractability with realistic assessment of counterfactual validity under potential distributional changes. In summary, Lemma 2.1 shows that with probability $\alpha = 1 - R^N$, a candidate counterfactual $x'$ remains valid under at least a fraction $R$ of plausible model realizations, thus providing a probabilistic robustness guarantee.

**Discussion**   Our method improves upon direct model parameter-bounding approaches. Deriving a tight parameter bound for deep neural networks is difficult as it requires analysing the global properties of non-convex training dynamics across all possible data distribution shifts, an unsolved challenge. By formulating robustness in terms of Wasserstein-bounded shifts $P_{\text{new}} \in \mathcal{B}_{\varepsilon_x}(\hat{\mathbb{P}}_{\text{old}})$, domain experts can explore probable perturbations with data constraints. Our framework is built on top of standard model training procedures deriving bounds naturally from the Wasserstein radius $\varepsilon_x$ and fine-tuning dynamics, making guarantees more flexible. Empirically, counterfactuals generated in this way exhibit lower $\ell_1$ proximity costs, and our probabilistic formulation (Lemma 2.1) further enables risk calibration through thresholds $(R, \alpha)$. In summary, the framework can be viewed as a *data-driven generalization of model parameter bounding*, ensuring robust CEs while providing additional flexibility and control over plausible shifts.

# 3 Experiments

The following sections first introduce the experimental setup, including evaluation, data, and competitors, followed by the presentation of our results. For more details, refer to Appendix A.2.

*Evaluation* We evaluate the quality of our generated counterfactuals regarding validity, proximity and plausibility. For validity we measure the validity of the base model (**v1**) and the shifted model (**v2**), also known as *Validity after Retraining*[Jiang et al., 2024]. Proximity is evaluated with $\ell_1$-distance and plausibility is assessed with Local Outlier Factor (**LOF**) [Breunig et al., 2000], that quantifies the outlierness of point, thereby assessing its closeness to the data manifold.

*Data* We consider four datasets: *Credit* [Dua and Graff, 2019], *Small Business Administration (SBA)* [Li et al., 2018], *Diabetes* [Smith et al., 1988], and *Adult* [Dua and Graff, 2019]. The first two datasets exhibit known distribution shifts [Upadhyay et al., 2021].

*Competitors* We primarily compare our method to Jiang et al. [2023] (referred to as RF), providing a worst-case guarantee for robust CEs against model changes and AP$\Delta$S [Marzari et al., 2024], a probabilistic framework designed to ensure robustness across probable model changes. We set $\alpha = 0.999$ and $R = 0.995$, yielding 1,378 realizations for robustness testing. To search for CEs within the robustness frameworks of RF, AP$\Delta$S, and our method, we evaluate the following state-of-the-art algorithms: *Proto* [Van Looveren and Klaise, 2021], which generates interpretable CEs by guiding the search via gradient descent with class prototypes, and *MILP* [Mohammadi et al., 2021], based on mixed-integer linear programming. Finally, we also include *ROAR* [Upadhyay et al., 2021], which enhances algorithmic recourse robustness using an adversarial training-inspired approach.

**Experimental Evaluation** Our method demonstrates comparable validity (**v2**) to RF and AP$\Delta S$ while achieving the best performance in terms of proximity ($\ell_1$). Both RF and AP$\Delta S$ guarantee the maximal model change, $\delta_{max}$. However, the use of Interval Neural Networks (INNs) in RF and AP$\Delta S$ can lead to both overestimation and underestimation of robustness, depending on how $\delta_{max}$ is defined. This, in turn, may result in misleading robustness guarantees. Rather than constraining the model parameters, we focus on modeling potential data shifts. In particular, we set $\varepsilon_x$ based on historical data (see Appendix A.2), which eliminates the need to constrain model parameter changes or perform a complex $\delta_{\max}$ search, while still providing an effective and flexible guidance mechanism for CE searches. As a result, our method achieves improved proximity, leading to better (lower) $\ell_1$ scores. Code is available at `https://github.com/zhaoxuan00707/robustness_ce`.

| | Diabetes | | | | Adult | | | | SBA | | | | Credit | | | |
|---|---|---|---|---|---|---|---|---|---|---|---|---|---|---|---|---|
| | **v1** | **v2** | $\ell_1$ | **lof** | **v1** | **v2** | $\ell_1$ | **lof** | **v1** | **v2** | $\ell_1$ | **lof** | **v1** | **v2** | $\ell_1$ | **lof** |
| Proto-AP$\Delta$S | 100% | 95% | 0.063 | 1.00 | 100% | 95% | 0.036 | 1.00 | 90% | 85% | 0.008 | 0.60 | 100% | 96% | 0.112 | -1.00 |
| Proto-RF | 100% | 96% | 0.104 | 1.00 | 100% | 100% | 0.069 | 1.00 | 90% | 88% | 0.011 | -0.02 | 82% | 80% | 0.300 | -1.00 |
| **Proto-Ours** | 100% | 96% | **0.024** | 1.00 | 100% | 100% | **0.019** | 1.00 | 90% | 88% | **0.007** | 0.53 | 92% | 90% | **0.019** | -1.00 |
| MILP-AP$\Delta$S | 100% | 94% | 0.049 | 0.96 | 100% | 91% | 0.032 | 1.00 | 92% | 81% | 0.007 | 0.56 | 91% | 85% | 0.024 | 1.00 |
| MILP-RF | 100% | 100% | 0.212 | -0.48 | 100% | 97% | 0.059 | 1.00 | 90% | 91% | 0.018 | -0.57 | 91% | 90% | 0.031 | 1.00 |
| **MILP-Ours** | 100% | 100% | **0.021** | 0.80 | 100% | 100% | **0.023** | 1.00 | 100% | 97% | **0.006** | -0.88 | 100% | 93% | **0.017** | 1.00 |
| ROAR | 82% | 40% | 0.076 | 0.95 | 78% | 79% | 0.071 | 1.00 | 82% | 79% | 0.035 | -0.80 | 62% | 55% | 0.027 | 1.00 |

Table 1: Performance comparison of different methods across datasets.

# 4 Conclusion and Future Work

We introduced a data-driven framework for counterfactual robustness under distributional uncertainty, leveraging linearized Wasserstein perturbations to provide probabilistic robustness guarantees for counterfactual explanations. Our results show that this approach generalizes classical parameter-bounding techniques while offering greater flexibility. By directly modeling plausible data shifts instead of constraining model parameters, we eliminate the complex $\delta_{\max}$ search, provide more faithful robustness guarantees, and achieve improved proximity.

Future works can target incorporating richer models of distributional change, such as adversarial perturbations, demographic or subpopulation shifts, and temporally evolving datasets, would enable finer-grained and more realistic robustness guarantees.

**Acknowledgement**  This work was partially funded by project W2/W3-108 Initiative and Networking Fund of the Helmholtz Association.

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

# A  Appendix

## A.1  Proof

Let the original (empirical) regularized objective be $R_{\text{old}}(\theta) := \frac{1}{n} \sum_{j=1}^{n} \ell(x_j, y_j; \theta) + \frac{\gamma}{2} \|\theta\|_2^2$, and denote its minimizer by $\theta_{\text{old}} := \arg\min_\theta R_{\text{old}}(\theta)$. Let the perturbed inputs be $x_j^{\text{new}} = x_j + \delta_j$ and define the perturbed empirical objective $R_{\text{new}}(\theta) := \frac{1}{n} \sum_{j=1}^{n} \ell(x_j + \delta_j, y_j; \theta) + \frac{\gamma}{2} \|\theta\|_2^2$, with minimizer $\theta_{\text{new}} := \arg\min_\theta R_{\text{new}}(\theta)$.

We work under the *linearized Wasserstein (squared) constraint*

$$\frac{1}{n} \sum_{j=1}^{n} c(x_j, x_j + \delta_j) = \frac{1}{n} \sum_{j=1}^{n} \|\delta_j\|_2^2 \le \varepsilon_x,$$

i.e., the ground cost is $c(x, x') = \|x - x'\|_2^2$ and $\varepsilon_x$ bounds the mean squared displacement.

**Assumptions.**

1. The per-sample loss $\ell(x, y; \theta)$ is differentiable in $\theta$.
2. The regularization parameter satisfies $\gamma > 0$, so $R_{\text{new}}$ and $R_{\text{old}}$ are $\gamma$-strongly convex in $\theta$.
3. The parameter gradient is Lipschitz in the input: there exists $L_x > 0$ such that for all $x, x', y, \theta$,
$$\|\nabla_\theta \ell(x', y; \theta) - \nabla_\theta \ell(x, y; \theta)\|_2 \le L_x \|x' - x\|_2.$$

**Theorem 1.** *Under the above assumptions, $\|\theta_{\text{new}} - \theta_{\text{old}}\|_2 \le \frac{L_x}{\gamma} \sqrt{\varepsilon_x}$.*

*Proof.* By first-order optimality of the minimizers we have
$$\nabla R_{\text{old}}(\theta_{\text{old}}) = 0, \qquad \nabla R_{\text{new}}(\theta_{\text{new}}) = 0.$$

Consider the function $R_{\text{new}}$. Strong convexity of $R_{\text{new}}$ implies the monotonicity inequality
$$\langle \nabla R_{\text{new}}(\theta_{\text{new}}) - \nabla R_{\text{new}}(\theta_{\text{old}}), \ \theta_{\text{new}} - \theta_{\text{old}} \rangle \ge \gamma \|\theta_{\text{new}} - \theta_{\text{old}}\|_2^2.$$

Using $\nabla R_{\text{new}}(\theta_{\text{new}}) = 0$ this becomes
$$-\langle \nabla R_{\text{new}}(\theta_{\text{old}}), \ \theta_{\text{new}} - \theta_{\text{old}} \rangle \ge \gamma \|\theta_{\text{new}} - \theta_{\text{old}}\|_2^2.$$

Applying Cauchy–Schwarz yields
$$\|\nabla R_{\text{new}}(\theta_{\text{old}})\|_2 \cdot \|\theta_{\text{new}} - \theta_{\text{old}}\|_2 \ge \gamma \|\theta_{\text{new}} - \theta_{\text{old}}\|_2^2.$$

If $\theta_{\text{new}} \ne \theta_{\text{old}}$ we can divide both sides by $\|\theta_{\text{new}} - \theta_{\text{old}}\|_2$ to obtain the basic stability inequality
$$\|\theta_{\text{new}} - \theta_{\text{old}}\|_2 \le \frac{1}{\gamma} \|\nabla R_{\text{new}}(\theta_{\text{old}})\|_2.$$

Using $\nabla R_{\text{old}}(\theta_{\text{old}}) = 0$ we rewrite the right-hand side as
$$\|\nabla R_{\text{new}}(\theta_{\text{old}})\|_2 = \big\|\nabla R_{\text{new}}(\theta_{\text{old}}) - \nabla R_{\text{old}}(\theta_{\text{old}})\big\|_2.$$

Expanding the gradients of the empirical risks gives
$$\big\|\nabla R_{\text{new}}(\theta_{\text{old}}) - \nabla R_{\text{old}}(\theta_{\text{old}})\big\|_2$$
$$= \bigg\| \frac{1}{n} \sum_{j=1}^{n} \big(\nabla_\theta \ell(x_j + \delta_j, y_j; \theta_{\text{old}}) - \nabla_\theta \ell(x_j, y_j; \theta_{\text{old}})\big) \bigg\|_2.$$

Applying the triangle inequality followed by the Lipschitz assumption yields
$$\big\|\nabla R_{\text{new}}(\theta_{\text{old}}) - \nabla R_{\text{old}}(\theta_{\text{old}})\big\|_2 \le \frac{1}{n} \sum_{j=1}^{n} \big\|\nabla_\theta \ell(x_j + \delta_j, y_j; \theta_{\text{old}}) - \nabla_\theta \ell(x_j, y_j; \theta_{\text{old}})\big\|_2$$
$$\le \frac{1}{n} \sum_{j=1}^{n} L_x \|\delta_j\|_2 = L_x \cdot \frac{1}{n} \sum_{j=1}^{n} \|\delta_j\|_2.$$

Finally, by Cauchy–Schwarz (or RMS–AM inequality),

$$\frac{1}{n}\sum_{j=1}^{n}\|\delta_j\|_2 \le \sqrt{\frac{1}{n}\sum_{j=1}^{n}\|\delta_j\|_2^2} \le \sqrt{\varepsilon_x},$$

where we used the linearized Wasserstein constraint $\frac{1}{n}\sum_j \|\delta_j\|_2^2 \le \varepsilon_x$. Combining the last three displays gives the desired bound

$$\|\theta_{\text{new}} - \theta_{\text{old}}\|_2 \le \frac{L_x}{\gamma}\sqrt{\varepsilon_x}.$$

$\square$

## A.2 Experimental Details

The networks in our experiments are built with Pytorch [Paszke et al., 2019]. Our networks are trained on an Intel(r) Core(TM) i7-8700 CPU.The iterative procedure of Algorithm 2 generates CEs of increasing distance until the target robustness is satisfied. For Proto, the distance of CEs is increased by iteratively amplifying the influence of the loss term related to CE validity. For MILP, the probability of the classifier's output for subsequent CEs is required to increase at each iteration (all test instances are classified as class 0, with the desired class being class 1).

|  | Diabetes | Adult | SBA | Credit |
| --- | --- | --- | --- | --- |
| AP$\Delta S$ | 12.45 | 23.67 | 35.89 | 47.12 |
| Ours | 1.23 | 2.56 | 3.78 | 4.90 |

Table 2: Average time (s) required to generate one CE using both MILP-AP$\Delta S$ and MILP-Ours.

To ensure fair comparability, weight decay is applied in all training processes. Table 2 compares the efficiency of CE generation time (in seconds) for two methods. While AP$\Delta S$ estimates $\delta_{max}$ more accurately than RF [Marzari et al., 2024], it requires searching for a distinct $\delta_{max}$ for each CE $x'$, which substantially reduces efficiency. In contrast, our method improves computational efficiency.

To introduce data shifts, we randomly shuffle the dataset and split it into two equal halves, denoted as $\mathcal{D}_1$ and $\mathcal{D}_2$. The base model (Feed Forward Neural Network) is trained on $\mathcal{D}_1$, while model updates are simulated through incremental retraining on $\mathcal{D}_2$, mimicking future model updates due to data shifts. In Jiang et al. [2023], fine-tuning is performed once a certain proportion (10%) of $\mathcal{D}_2$ is available via random sampling. In contrast, we use the newly arrived portion of $\mathcal{D}_2$ to estimate the value of $\varepsilon_x$ by calculating the Wasserstein distance between datasets. To evaluate the robustness of CEs, we consider two validity metrics: **v1** – the percentage of CEs that remain valid on the original base model; **v2** – the percentage of CEs that remain valid after the model is retrained on both $\mathcal{D}_1$ and $\mathcal{D}_2$. For each dataset, we generate 100 CEs and assess their quality based on: Proximity – measured using the $\ell_1$ distance and Plausibility – measured using the local outlier factor (**lof**), which evaluates whether an instance lies within the data manifold by analyzing local density [Breunig et al., 2000] (+1 for inliers, -1 otherwise). The $\ell_1$ distance and **lof** scores are averaged over the generated CEs, and the experimental results are summarized in Table 1.

In this initial work, we employ Gaussian feature perturbation for its simplicity and computational efficiency, acknowledging it as a first-step approximation of data shift. Future works will explore more sophisticated, semantically-grounded perturbation methods.

