# OpenReview forum: "Probabilistic Framework for Robustness of Counterfactual Explanations Under Data Shifts"
_NeurIPS.cc/2025/Workshop/Reliable_ML — NeurIPS 2025 - Reliable ML Workshop_

### Official Review · Reviewer_5pUi · 2025-09-18
**Well-motivated and appropriate paper with mathematical details to fix**

**Rating:** 7
**Confidence:** 4

**Review:**

# Summary. What the paper claims, how it does it, and the main results.

This paper introduces a novel method for ensuring that counterfactual explanations (CEs) remain robust even under potential model shift resulting from underlying data distributional shift. Concretely, the method first samples data distributions from an approximate Wasserstein ball around the current distribution, then performs a limited amount of gradient descent on each new distribution to produce perturbed model parameters. The CE is then tested for validity under each of these perturbed model parameters (e.g., whether or not the probability of the desired label is above a decision threshold), thereby (through a lemma) producing a confidence interval on the percentage of future model shifts under which the CE will remain valid. Theoretical results also show that a Wasserstein data bound produces a hypersphere bound on model parameter shift. Empirical results show favorable performance relative to existing model-shift-bounding methods.

# Strengths. Novelty, rigor, empirical/theoretical quality, clarity, relevance to reliability with imperfect data.

* The idea is novel and well-motivated.
* The application is fairly well-suited to this workshop.
* The theory is interesting and thought-provoking.

# Weaknesses / Limitations. Missing comparisons/ablations, unclear assumptions, proof gaps, failure modes, scope limits.

* Lemma 2.1 seems incorrect as stated, though it is not hard to figure out what it is trying to say. Note, for example, that the min_i on the right side of the inequalities is currently meaningless; there is only one theta_old. Really, it should be a min over the sampled theta_new^(i) on the right side. Then, if you want to hew exactly to Marzari et al. (2024), the left side should be a min over future sequences of thetas; however, that presentation is also mathematically sketchy and wrong as-stated. It would be far clearer to formally write the statement that you later interpret the lemma as saying: that (parentheses for parsing) the probability that (the probability of any theta drawn from the future distribution of thetas being above the min is at least R) is at least alpha. This is the Wilks result, so you could just cite Wilks (1942; see Marzari paper for exact citation) and cite Marzari et al. (2024) as inspiration.

(My score is contingent on the lemma's statement being fixed; it's not understandable on its own as is.)
* The notation around f(x) is also confusing and arguably a little bit sloppy. In the problem setup, it’s considered to be a discrete category output: f(x) = y. But then a min is taken over f(x) in Lemma 2.1; it’s not clear what this means. Presumably, the intention in practice is for this to mean the logits of y outputted by f given x, but this currently needs to be inferred by the reader and causes confusion.

# Suggestions for Authors. Specific things that would improve the paper:

* Fix the weaknesses above.
* The purpose of Section 2.1.1 and its connection (or lack thereof) to Lemma 2.1 could be clearer. I think it would be easier to follow if you put Section 2.1.1 first as a way of setting the stage: “this is what happens to the model parameters when you perturb within a Wasserstein ball on the data, and this is how you can perform inference given those perturbed model parameters.”
* Experiment setup is too long to parse right now. It can be its own subsection with multiple paragraphs.
* Similarly, second related works paragraph is far too long. Make it a subsection with a \paragraph{} title.

# Ethics (if applicable). Note any concerns (about privacy, fairness, misuse, sensitive data use) and suggested mitigations.
N/A

---

### Official Review · Reviewer_awWP · 2025-09-18
**Interesting paper on counterfactual robustness, more model justification needed.**

**Rating:** 7
**Confidence:** 3

**Review:**

**Summary:**
- This paper provides a data-driven method to measure counterfactual robustness under linearized Wasserstein perturbations. More precisely, they assume there is some initial model parameterized by a value $\theta$ trained on an uncorrupted dataset X. This dataset then goes under a Wasserstein shift to yield a new dataset X’. They then quantify how robust counterfactual explanations are to this shift.

**Strengths:**
- The authors motivate well the need for counterfactual explanations and the existing literature.
- The guarantees provided seem to offer more flexibility compared to a worst-case parameter shift guarantee.
- The authors provide nice experimental results showing that this method outperforms $Proto-AP\Delta S$ and $Proto-RF$.

**Weaknesses:**
- Lines 110-121 feel very crowded and repeat a proof found in the appendix. I would either include the full proof in the main body or just reference the appendix and provide a brief description of the proof strategy in words.
- The results seem to rely heavily on Lemma 2.1 from Marzari et al 2024.
- I would like more motivation on why Wasserstein distance is the correct metric to use for counterfactual explanations.

**Suggestions:**
- It’s not clear to me why using this Monte Carlo method is better or more flexible than considering parameter bounding. In practice, how would someone generate the delta_j perturbations?
- I would like more discussion about why we would not want a worst-case perturbation type result. Why is this data-driven method actually better in practice?
- When you reference the appendix, please put what section you are referencing.